# MicroRNAs and Progesterone Receptor Signaling in Endometriosis Pathophysiology

**DOI:** 10.3390/cells11071096

**Published:** 2022-03-24

**Authors:** Warren B. Nothnick

**Affiliations:** 1Center for Reproductive Sciences, Department of Molecular and Integrative Physiology, Kansas City, KS 66160, USA; wnothnic@kumc.edu; Tel.: +1-(913)-588-6277; 2Department of Obstetrics and Gynecology, University of Kansas Medical Center, Kansas City, KS 66160, USA; 3Institute for Reproduction and Perinatal Research, Center for Reproductive Sciences, University of Kansas Medical Center, Kansas City, KS 66160, USA

**Keywords:** progesterone receptor, progesterone resistance, microRNA, endometriosis

## Abstract

Endometriosis is a significant disease characterized by infertility and pelvic pain in which endometrial stromal and glandular tissue grow in ectopic locations. Altered responsiveness to progesterone is a contributing factor to endometriosis pathophysiology, but the precise mechanisms are poorly understood. Progesterone resistance influences both the eutopic and ectopic (endometriotic lesion) endometrium. An inability of the eutopic endometrium to properly respond to progesterone is believed to contribute to the infertility associated with the disease, while an altered responsiveness of endometriotic lesion tissue may contribute to the survival of the ectopic tissue and associated symptoms. Women with endometriosis express altered levels of several endometrial progesterone target genes which may be due to the abnormal expression and/or function of progesterone receptors and/or chaperone proteins, as well as inflammation, genetics, and epigenetics. MiRNAs are a class of epigenetic modulators proposed to play a role in endometriosis pathophysiology, including the modulation of progesterone signaling. In this paper, we summarize the role of progesterone receptors and progesterone signaling in endometriosis pathophysiology, review miRNAs, which are over-expressed in endometriosis tissues and fluids, and follow this with a discussion on the potential regulation of key progesterone signaling components by these miRNAs, concluding with suggestions for future research endeavors in this area.

## 1. Endometriosis: Steroid Dependency and Resistance

Endometriosis is a common disease in women of reproductive age characterized by pain, dysmenorrhea, and infertility, in addition to a serious impairment of their ability to manage daily activities [1]. Endometriosis is defined as the growth of endometrial stromal and glandular tissue in ectopic locations, with lesions predominantly developing within the pelvic cavity on the surface of the peritoneal lining and ovaries as well as the cul-de-sac region [1,2,3]. It is well-established that endometriosis is a disease of estrogen dependence, and that reducing estrogen action and/or production is an effective means for treating the disease. Of these anti-estrogen therapies, progesterone analogs are widely used. Combined oral contraceptives which contain progesterone as well as progesterone-containing intra-uterine devices (IUDs) are common therapies for the disease [4,5]. Unfortunately, progesterone analogs are not effective in all endometriosis patients, and this is proposed to be due to progesterone resistance associated with the disease [5]. 

Evidence of progesterone resistance is not only manifested in lack of efficacy of the steroid to reduce pain and other symptomology of the disease, which is presumed to result from the persistence of ectopic lesion tissue, but also from research studies which have demonstrated impaired progesterone signaling in the eutopic endometrial tissue of patients with endometriosis. Several studies have reported the altered expression of progesterone target genes in both eutopic endometrium and endometriotic lesion tissue. Taylor and colleagues [6] first reported that patients with endometriosis failed to show the anticipated increase in *HOXA10* and *HOXA11* gene expression in mid-secretory stage endometrial tissue. Expression profiling of endometrium from women with endometriosis revealed candidate genes for endometriosis-associated implantation failure and infertility [7] as well as progesterone resistance [8]. Many of these mis-expressed genes are known progesterone targets which play an essential role in endometrial physiology and function. Endometriotic lesion tissue also displays progesterone resistance and altered gene expression [9], and this is proposed to be at least in part due to altered progesterone receptor (PGR) expression. Mechanistically, studies suggest that this misexpression may be due to altered AKT and/or MEK 1/2 activity [10], the inflammatory milieu associated with the disease [11], and/or epithelial-to-mesenchymal transition [12]. Taken together, both eutopic and ectopic endometrial tissue exhibit altered progesterone responsiveness, and this may in part be due to the altered expression of PGRs. However, limiting the assessment to only the classic PGRs provides only a partial understanding of progesterone signaling. In the following paragraphs, we summarize what is known about the classic, genomic receptors as well as the non-genomic receptors. 

## 2. Progesterone Receptors and Coactivators in Endometrial Physiology and Endometriosis Pathophysiology

Within the endometrium, progesterone regulates the expression of specific genes predominantly through the activation of the cognate nuclear PGRs. Two distinct forms of the PGR have been identified which are produced by the transcription of two separate mRNAs, both from the same gene [13]. PGR-A is a truncated form of PGR-B, lacking the N-terminal 164 amino acids present in PGR-B progesterone receptors, but both exhibit identical DNA-binding and steroid binding domains [14,15,16,17]. PGR-A and PGR-B are both expressed in the endometrial glandular epithelium, with PGR-B dominance occurring during the midsecretory stage of the menstrual cycle, while PGR-A exhibits predominance within stroma cells throughout the menstrual cycle [18]. Collectively, these data suggest that PGR-A and PGR-B mediate distinct pathways of progesterone action in the glandular epithelium and stroma during the menstrual cycle. More recently, nongenomic mechanisms for progesterone signaling have been reported, including membrane progesterone receptors (mPRs) that belong to the class II members of the progesterone and adipoQ receptor (PAQR) family as well as progesterone receptor membrane components (PGRMCs). The following paragraphs summarize what is known about these different receptor signaling pathways in the pathophysiology and progesterone resistance in endometriosis in both eutopic endometrial tissue and ectopic lesion tissue.

### 2.1. Classic/Genomic PGRs in Endometrial Physiology and Endometriosis Pathophysiology

The classical, genomic pathway for progesterone modulation of endometrial genes is mediated by two isoforms of the PGR, PGR-A and PGR-B. In the human endometrium, PGR-A is primarily expressed in stromal cells, while PGR-B is the predominant isoform in epithelial cells with faint expression in stromal cells [18,19], suggesting that PGR-A may be the primary mediator of progesterone action in endometrial tissue.

Assessment of PGR expression in endometriosis ectopic lesion tissue and eutopic endometrial tissue had produced conflicting results. Independent studies by Lyndrup and colleagues [20] and Jones and associates [21] reported lower PGR levels in ectopic lesion tissue compared with eutopic endometrium. In contrast, a study by Nisolle and coworkers reported no difference in expression [22], while Lessey and colleagues [23] reported that in endometriotic lesion tissue, PGR expression was more heterogeneous and did not undergo predictable changes in response to endogenous hormones. It should be noted that in these studies, the employed methodology did not allow for discrimination between PGR-A and PGR-B, leaving the possibility that the ratio of PGR-A/PGR-B could change, which in turn could alter progesterone responsiveness. A latter study by Attia and associates [24] used Western blots to distinguish between PGR isoforms and reported that peritoneal endometriotic lesion tissue did not express PGR-B and exhibited lower levels of PGR-A compared with the eutopic endometrium. In contrast, Misao and coworkers [25] reported the dominant expression of *PGR-B* transcript in ovarian endometriotic tissue compared to the eutopic endometrium. 

Studies of PGR-A and PGR-B levels in the eutopic endometrium in women with and without endometriosis produced similar conflicting outcomes. Bedaiwy and associates [26] reported that eutopic endometrial PGR-A expression was significantly elevated in women with endometriosis compared to disease free controls, and this elevation occurred independent of the stage of the menstrual cycle. Furthermore, PGR-A levels were also significantly elevated in ovarian endometriotic lesions compared to peritoneal lesions. In accord with these findings, the assessment of progesterone responsive genes in eutopic endometrial tissue from women with endometriosis also revealed that the PGR transcript was increased compared to controls [8]. In contrast, eutopic endometrial tissue expression of PGR was reported to show no differences between endometriosis and control subjects, but lower expression in both glands and stroma of the ectopic lesion [27], while assessment of PGR transcript expression in eutopic endometrial tissue from women with endometriosis failed to exhibit differences in expression level compared to eutopic endometrium from infertile and fertile control subjects [28]. Thus, for both eutopic and ectopic lesion tissue, results are conflicting with respect to the expression of PGR despite the reported progesterone resistance characteristic of this tissue. These observations raise the possibility that other progesterone signaling pathways may contribute to the insensitivity of this steroid in endometriosis pathophysiology. 

### 2.2. mPRs/PAQRs in Endometrial Physiology and Endometriosis Pathophysiology

While the majority of research on progesterone resistance and endometriosis has focused on the ‘classic’, genomic pathways for progesterone signaling, progesterone also has rapid, non-genomic action effects on cell-signaling pathways independently of transcription [29]. Non-genomic actions of progesterone appear to be mediated at the level of the cell membrane, as tissues which do not express classical PGRs exhibit progesterone responsiveness. Further, in vitro studies which coupled the steroid with bovine serum albumin, rendering it incapable of crossing the plasma membrane and interacting with classical PGRs, support the notion of membrane receptors capable of eliciting progesterone signaling [30,31,32].

Zhu and associates [33] were the first to identify a set of putative membrane PGRs which were named mPRα (or *PAQRVII*/*PAQR7*), mPRβ (or *PAQRVIII*/*PAQR8*) and mPRγ (or *PAQRV*/*PAQR5*). These receptors mediate non-genomic progesterone signaling acting as G protein-coupled receptors (GPCRs) [33,34,35]. Within the human endometrium, mPRα appears to be the predominant membrane PGR [36]. Transcripts for mPRα, mPRβ, and mPRγ as well as *PAQRIII* and *PAQRIX* were all expressed in human endometrium across the menstrual cycle, but only mPRα, mPRγ and *PAQRIX* expression varied across the menstrual cycle. More specifically, mPRα expression rose during the early and late secretory stages of the cycle compared to early/mid-proliferative stages, which coincides with elevated progesterone levels. In contrast, both mPRγ and *PAQRIX* transcript expression were lower during the early and late secretory stages. From this, the authors concluded that mPRα may mediate progesterone action in tissues that express low levels of PGR. More recently, Sinreih and colleagues used immunohistochemical localization techniques to define the spatial expression of mPRs in endometrial tissues. mPRα and mPRβ were localized primarily to cell membranes, while mPRγ was localized in the cytoplasm and/or nucleus of normal endometrial tissue, while mPRα and mPRβ were detected at the cell membrane or in the cytoplasm, or both, while mPRγ was only localized in the cytoplasm of endometrial cancer tissue [37]. Within the context of endometriosis, mPRs expression was recently reported for both transcript and protein [38]. mPRα and mPRγ transcript expression (*PAQR7* and *PAQR8*, respectively) were both lower in eutopic endometrium and ectopic lesion tissue compared to control endometrial tissue, while transcript expression of mPRβ (*PAQR8*) and mPRδ (*PAQR6*) was only reduced in the eutopic endometrium, with lesion tissue expressing similar levels to that of the control endometrium. Protein expression of mPRα and mPRβ was decreased in the ectopic lesion tissue while expression of mPRδ tended to be lower in ectopic tissue (P>0.05). These findings can be interpreted to suggest that low levels of these mPRs may also contribute to endometriosis-associated progesterone resistance. In summary, mPRs are expressed in menstrual cycle-stage specific patterns and levels are reduced in both eutopic and ectopic endometriosis tissue; however, the mechanisms which lead to this reduction have not been studied. 

### 2.3. PGRMCs in Endometrial Physiology and Endometriosis Pathophysiology

In addition to the aforementioned membrane progesterone receptors which are GCPRs, two other proteins with progestin binding activity have been identified and are referred to as progesterone receptor membrane component (PGRMC) 1 and PGRMC2. In uterine tissue, PGRMC1 and PGRMC2 were first characterized in mice [39]. During the estrous cycle, *Pgrmc2* transcript levels were highest during proestrus and metestrus, while *Pgrmc1* levels remained constant across the estrous cycle. Pgrmc1 protein was predominantly expressed in glandular epithelial cells during proestrus, with stromal cell upregulation from estrus to metestrus, declining during diestrus. Steroid reconstitution studies using ovariectomized mice [39] demonstrated that estrogen alone or in combination with progesterone had minimal effect on Pgrmc1 protein expression, while progesterone alone induced a significant increase in its expression. 

To further evaluate the role of these PGRMCs in endometrial/uterine function, Pru and Clark [40] reported that Pgrmc1 deficiency in the murine endometrium results in the development of cystic glandular hyperplasia with incidence increasing with increasing age. Associated with this phenotype is an increase in epithelial cell apoptosis, the infiltration of immune cells, heavily vacuolated epithelial cells, and disruption of the transitional zone between the epithelium and the underlying stromal tissue. Due to the fact that Pgrmc1 was deleted from the stroma/mesenchyme (*Amhr2-Cre* mice were used to delete Pgrmc1), these observations would suggest that paracrine signaling between the stromal and epithelial compartments is disrupted in Pgrmc1 cKO mice. The hyperplastic characteristics of uterine/endometrial tissue of these mice would also suggest unopposed estrogen/reduced progesterone signaling. 

Similar studies by the Pru laboratory evaluated the role of Pgrmc2 and Pgrmc1/2 using mice in which Pgrmc2 or both Pgrmcs were conditionally deleted from uterine tissue [41]. For this study, Pgrmc1 and/or Pgrmc2 floxed mice (Pgrmc2fl/fl and Pgrmc1/2fl/fl) were crossed with *Pgr-cre* mice, resulting in the conditional ablation of Pgrmc1 and/or Pgrmc2 from female reproductive tissues including the uterus/endometrium. Ablation of Pgrmc2 initially led to subfertility and reproductive senescence, with double deletion of Pgrmc1/2 yielding a similar phenotype. In both models, the subfertility was due to postimplantation embryonic death independent of ovarian insufficiencies as well as cystic endometrial glands similar to those observed in their prior study with Pgrmc1 conditionally deleted mice. 

With respect to endometriosis, both PGRMC1 and PGRMC2 have been evaluated. Using a non-human primate model, Keator and colleagues [42] defined *PGRMC1* and *PGRMC2* transcript expression levels in well-defined samples of endometrium collected from artificially cycled macaques during the menstrual cycle, and in the secretory phase endometrium of naturally cycling macaques afflicted with endometriosis. PGRMC1 and PGR were elevated during the proliferative phases of the cycle, and then declined to nearly undetectable levels during the late secretory phase of the cycle. Levels of PGRMC2 were lowest during the proliferative phases of the cycle and then increased markedly during the secretory phases. PGRMC2 protein predominantly localized to luminal and glandular epithelia during the secretory phases. Compared to controls, macaques with endometriosis exhibited no changes in the expression or localization patterns for PGR and PGRMC1, but exhibited strikingly reduced levels of PGRMC2 transcript and altered intracellular staining patterns for the PGRMC2 protein. Collectively, these results suggest that membrane-bound PGRMC2 may provide a pathway of action that could potentially mediate the non-genomic effects of progesterone on the glandular epithelia during the secretory phase of the cycle. Furthermore, reduced levels of membrane-bound PGRMC2 may be associated with the progesterone insensitivity often observed in the endometrium of primates afflicted with endometriosis.

Utilizing human eutopic endometrial tissue from women with severe (stage III/IV) endometriosis and disease-free women, Bunch and associates [43] found that *PGRMC1* and *PGRMC2* transcript expression levels were significantly downregulated in the secretory phase in the eutopic endometrium in women with endometriosis. Using immunohistochemical localization, it was further discovered that the decrease in both PGRMCs in the secretory phase endometrial tissue of endometriosis subjects was primary due to stroma cell deficiency. The observation of reduced PGRMC2 is in accord with those reported in a non-human primate model of endometriosis [42], while only humans with endometriosis appear to also exhibit reduced PGRMC1 expression. Nonetheless, PGRMCs may play a role in the eutopic endometrial progesterone resistance associated with endometriosis, but information with respect to lesion tissue expression is lacking. 

In summary, the expression of PGR-A and PGR-B as well as the mPRs (PAQRs) and PGRMCs is altered in eutopic and/or endometriotic lesion tissue from women with endometriosis, which presumptively contributes to the progesterone resistance characteristic of this disease. What is still poorly understood are the pathways which may lead to reduced expression of these progesterone receptors. One of the potential mediators for this misexpression may be the small, non-coding RNAs know as microRNAs (miRNAs).

## 3. Role of miRNAs in Endometriosis Progesterone Resistance

### 3.1. miRNA Biogenesis and Function

miRNAs play critical roles in gene expression [44,45], primarily via binding to miRNA response elements in the 3′ UTR of target mRNAs leading to translational repression [46,47]. These small non-coding RNAs are generated primarily through a canonical biogenesis pathway by which miRNAs are processed. miRNAs transcribed via this route rely on RNA polymerase II (Pol II) to generate pri-miRNAs, which contain an RNA hairpin in which one of the two strands includes the mature miRNA (Figure 1). Pri-miRNAs are then cleaved by DROSHA and DCG8, resulting in liberation of a precursor, or pre-miRNA. Pre-miRNAs are exported from the nucleus to the cytoplasm via exportin-5 (XPO5) for final processing by Dicer and TRBP, yielding a mature miRNA duplex which unwinds into single stranded mature miRNAs which are then incorporated into the RNA-induced silencing complex (RISC). The RISC then leads to the translational repression of proteins by either mRNA decay, or mRNA stabilization, which prevents translation. To a much lesser extent, some miRNA/RISC complexes can enhance translation (Figure 1). miRNAs can also regulate translation within the same cell or can be secreted into extracellular fluids/circulation. Secreted miRNAs can be contained within extracellular vesicles such as exosomes [48], or by forming complexes with proteins, such as AGO2 [49,50]. Secreted miRNAs can be taken up by recipient cells to regulate their activities and function as paracrine signaling mediators. For example, fibroblast-derived miR-21* was reported to exhibit paracrine signaling with cardiomyocytes [51], while endothelial-derived miR-423-5p induced Müller cell activation in diabetic retinopathy [52].

Considerable work in the field of miRNAs in both endometrial physiology and pathophysiology, including endometriosis, has been conducted over the last several years [53,54], but the specific role of miRNA regulation in mediating progesterone signaling/progesterone resistance is underexplored. The following paragraphs will discuss miRNAs which target the different PGRs as related to endometriosis pathophysiology. To identify validated as well as putative miRNAs which may regulate PGR (A/B), PGRMC1, PGRMC2, PAQR5, PAQR6, PAQR7 and PAQR8, we utilized two approaches. First, we examined the literature to identify miRNAs which had been reported to target progesterone receptors regardless of tissue/cell types. Secondly, we evaluated miRNAs which were reported in at least two separate studies to be either elevated in endometriotic tissue/cells and/or in the circulation (serum and/or plasma). From this list, we then searched the literature as well as utilized TargetScan [55], mirDIP [56] and miRDB [57] bioinformatic programs to identify if any of these miRNA have been validated (original research) or predicted (bioinformatics) to target the aforementioned PGR receptors in endometriotic eutopic and/or ectopic tissues/cells.

### 3.2. MicroRNAs Which Are Validated to Target PGRs

#### 3.2.1. PGR-A/B

Of the aforementioned progesterone receptors, only PGR-A/B and PGRMC1 have been evaluated for regulation by miRNAs and confirmed to be directly regulated by the binding of miRNA with the specific 3′ UTRs (Figure 2). The earliest studies on miRNA regulation of PGR-A/B expression were conducted in the field of breast cancer research. Cui and colleagues used 3′UTR luciferase reporter assays and western blotting to confirm miR-126-3p regulation of PGR expression as well as proliferation and β-casein expression of mouse mammary epithelial cells [58], while miR-129-2 was reported to down-regulate PGR expression in response to progesterone treatment in breast cancer cell lines [59]. Muti and associates [60] reported that miR-513-a-5p, miR-513b-5p and miR-513c-5p were among the most significantly deregulated miRNAs in candidates that developed breast cancer with miR-513a-5p upregulation directly associated with breast cancer risk. Furthermore, experimental data corroborated the inhibitory function of miR-513a-5p on progesterone.

Receptor expression, confirming it as a target of miR-513a-5p. miRNA profiling, was conducted on T47D breast cancer cells treated with medroxyprogesterone acetate (MPA; a progesterone analog), it and identified 28 miRNAs which were differentially expressed, with 20 decreasing and eight increasing in expression [61]. Many of these differentially expressed miRNAs putatively target progesterone-responsive genes as well as PGR itself. miR-513a-5p was up-regulated in response to MPA treatment and could decrease PGR expression, providing a negative feedback mechanism to control PGR expression. 

Additional studies have revealed that miR-378-3p regulates PGR expression in ovarian granulosa cells [62], while in non-human primate endometrial tissue, PGR is a valid target of miR-96 in rhesus monkeys as well as in humans but not in rodents, whereas the regulation of PGR by miR-375 is rhesus monkey-specific [63]. With respect to endometriosis, Zhou and colleagues [64] demonstrated that miR-196a targeted PGR expression in endometrial stromal cells. Of interest was the observation that although miR-196a is predicted to target *PGR*, 3′ UTR reporter assays failed to confirm miRNA binding. It was revealed that miR-196a’s down-regulation of PGR occurred indirectly via the modulation of MEK/ERK. miR-194-3p was also reported to be increased in the eutopic endometrium from women with endometriosis which was associated with reduced PGR levels [65]. Further, PGR was identified by luciferase reporter assays as a direct target of miR-194-3p and its overexpression in stromal cells inhibited in vitro decidualization.

#### 3.2.2. PGRMC1/2

PGRMC1/2 have also been shown to be regulated by miRNAs in several tissues. Two miRNA families, miRNA let-7/miR-98 and miR-200a/141, are predicted to target PGRMC1 with the 3′UTR containing one highly and one poorly conserved binding site for the former, and one highly conserved binding site for the later. Using luciferase reporter assays for 3′UTR binding, it was revealed that let-7i and miR-98 targeted PGRMC1 in SKOV-3 cells, but the conserved binding site for miR-200a/141 were not functional [66]. In rat granulosa cells, hyaluronic acid up-regulated PGRMC1 expression via suppression of miR-139-5p [67]. PGRMC1 was validated as a direct target of miR-139-5p using both reporter assays and western blots. In the well-differentiated endometrial adenocarcinoma cell line, Ishikawa cells, miR-98 gain of function repressed PGRMC1 expression and cell proliferation, and PGRMC1 was confirmed to be a direct target of this miRNA [68]. To date, one report identified miR-3687 as a modulator of PGRMC2 expression in human esophageal squamous cell carcinoma cells, but direct binding with the 3′UTR was not confirmed [69]. miR-139-5p is an intriguing miRNA whose expression was first reported to be increased in endometriotic lesion stromal cells in 2018 [70], with a more recent report confirming the elevation of miR-139-5p in endometriotic lesion tissue as well as stromal cells [71]. In the study by Rekker and coworkers [70], the overexpression of miR-139-5p induced a down-regulation of HOXA10, which is suppressed in the endometriosis eutopic endometrium [6,72], specifically within the stromal cells [72]. Thus, it is plausible that miR-139-5p may induce repressed HOXA10 expression through the down-regulation of PGRMC1 as well as at the level of the *HOXA10* 3′UTR which it is predicted to target [55].

### 3.3. miRNAs Which Are Increased in Endometriosis Tissue and Fluids Predicted to Target PGRs

miRNAs and their potential role in the pathophysiology of endometriosis has been extensively explored. Considering the well-accepted role of altered progesterone responsiveness in the disease, the lack of miRNA assessment of PGR expression is surprising. Eight miRNAs which putatively target the three classes of progesterone receptors have been reported to be elevated in lesion tissue/cells (indicated by the red arrow) and/or the circulation (serum, plasma indicated by the pale orange arrow) of women with endometriosis (Figure 3). These miRNAs are proposed to act in either an autocrine (from the lesion tissue) or paracrine/endocrine fashion (from the circulation). Of these eight miRNAs, three (highlighted in bold text in Figure 3) have been evaluated in the context of progesterone signaling, while the remaining five have not. The putative role of these miRNAs in modulating PGR expression, progesterone signaling and events conducive to the pathophysiology of endometriosis is discussed in the following paragraphs.

#### 3.3.1. Elevated miRNAs Which Promote Lesion Survival and Progesterone Resistance

Of the elevated miRNAs, three have been reported to modulate the expression of progesterone receptors as well as events conducive with progesterone resistance relevant to endometriosis pathophysiology. These include miRNAs miR-29c-3p, miR-126-3p, and miR-143-3p, the latter which is overexpressed in both the circulation and lesion tissue of women with endometriosis.

miR-29c has been reported by three separate laboratories to be elevated in endometriotic lesion tissue compared to the eutopic endometrium [73,74,75]. Of these three reports, the study by Hawkins and colleagues [75] was the only one to assess the potential function of a differentially expressed miRNA. miR-29c, which exhibited the highest degree of differential expression between tissue types, was evaluated using primary human endometrial stromal cells in vitro. Extracellular matrix (ECM) proteins, which are putative targets of miR-29c, including COL7A1, UPK1B, and TFAP2C, were downregulated in cells overexpressing miR-29c mimic. Furthermore, luciferase reporter assays confirmed a direct effect on the 3′-UTR of these genes. Thus, the mis-expression of miR-29c in ovarian endometriomas appears to functionally contribute to the aberrant expression of ECM proteins associated with the disease, at least in isolated endometrial stromal cell cultures.

More recently, Long and associates [76] also investigated the expression of the miR-29 family (miR-29a, miR-29b, and miR-29c) in the endometrium from women without endometriosis, as well as in paired ectopic and eutopic endometrial samples. As opposed to earlier reports, refs. [73,74,75] these authors observed a decrease in the lesion expression of miR-29c, but the type (peritoneal, ovarian endometrioma, etc.) of lesion was not specified. In vitro studies using the CRL-7566 endometriosis cell line in which miR-29c was upregulated was associated with reduced endometrial cell proliferation and invasion, promoting cell apoptosis, and this mechanism included the regulation of c-Jun. It should be emphasized that while these investigators demonstrated a decrease in miR-29c expression in endometriotic lesion tissue which would suggest enhanced cell proliferation and invasion due to low miR-29c expression, all of the earlier reports demonstrated an increase in miR-29c expression. 

Joshi and colleagues [77] evaluated the expression of miR-29c-3p (miR-29c) in the endometrium of humans and baboons with or without endometriosis. This study revealed that miR-29c expression increased, while mRNA levels of its predicted target, FK506-binding protein 4 (FKBP4), decreased in eutopic endometrial tissue from baboons after the induction of experimental endometriosis. Using decidualized human uterine fibroblast/stromal cells, the forced expression of miR-29c mimic lead to the reduced expression of decidualization markers as well as FKBP4. A similar elevated expression of miR-29c was observed in human eutopic endometrial tissue from patients with endometriosis compared to controls. *FKBP4* 3′ UTR contains a miR-29c-3p binding site which is broadly conserved among vertebrates, while the *PGR* 3′ UTR contains a single miR-29c-3p binding site which is poorly conserved across vertebrates but is conserved between humans, chimpanzees and rhesus macaques [55]. Thus, miR-29c-3p could potentially target both the PGR as well as an important chaperone protein (FKBP52/FKBP4) required for P4 signaling. Collectively, the majority of the studies which have evaluated miR-29c-3p reported data that would suggest that elevated levels of miR-29c-3p in endometriosis could reduce progesterone signaling and lead to endometriotic lesion tissue survival; however, the direct effect of this miRNA on progesterone receptor expression has not been examined.

miR-126-3p was initially identified as a metastasis suppressing miRNA whose expression is down-regulated in relapsing breast cancer, leukemia and cervical cancer [78,79,80]. Subsequently, using mammary/breast tissue and cells, miR-126-3p was the first miRNA demonstrated to directly target the 3′ UTR of the progesterone receptor, stabilizing transcript levels leading to reduced PGR protein expression and P4 signaling [58]. Cui and colleagues [58] demonstrated that miR-126-3p may play a role in mouse mammary gland development, as it targets and regulates PGR protein expression, modulates viability of mammary epithelial cells and their secretion of β-casein. miR-126-3p has also been reported to play a role in mouse mammary gland lipid metabolism [81]. Estradiol and progesterone increase MCF-10A (non-malignant breast epithelial cells) fatty acid synthesis, which has been associated with the significant reduction in miR-126-3p expression. Together, these data can be interpreted to suggest a potential E2/P4 regulation of miR-126-3p which contributes to the modulating of PGR and P4 signaling. 

miR-143-3p expression is elevated in endometriotic lesion tissue [73,74] compared to matched eutopic endometrial tissue. Cosar and colleagues [82] also reported that miR-143-3p is elevated in serum from women with endometriosis compared to women found to be disease free, while Papari and coworkers reported lower levels of miR-143-3p in plasma of women with the disease [83]. From a functional standpoint, data on miR-143-3p function is also conflicting. Yang and colleagues [84] reported elevated levels of expression in the endometriotic stromal cells miR-143-3p compared to controls. Further, the functional assessment of miR-143-3p overexpression inhibited endometrial stromal cell proliferation and invasion, while knockdown of miR-143-3p knockdown reversed these effects. The authors concluded from these results that elevated levels of miR-143-3p inhibited events conducive to endometriosis progression. While Lin and associates [85] also detected upregulated levels of miR-143-3p in endometriotic lesion tissue, in vitro data suggested a permissive role for this miRNA in endometriosis progression. Knockdown of miR-143-3p was associated with the suppression of epithelial-to-mesenchymal transition, invasion and migration of endometriotic stromal cells. These results may be interpreted to suggest that elevated levels of miR-143-3p characteristic of endometriotic lesion tissue/cells enhance cellular events such as motility and invasiveness which are conducive to the progression of the disease and are in agreement with progesterone resistance. In summary, elevated levels of miR-29c-3p, miR-126-3p and miR-143-3p appear to promote some of the cellular events conducive to endometriosis lesion survival and progression. Augmentation of these cellular events could be due to a lack of progesterone responsiveness in endometriotic and/or eutopic endometrial tissue manifested by the over-expression of these miRNAs. If true, the increased expression of miR-29c-3p, miR-126-3p and/or miR-143-3p may be causative factors in the development, progression and progesterone resistance of endometriosis. 

#### 3.3.2. Elevated miRNAs Which Putatively Target Progesterone Receptors but Suppress Lesion Survival

Of the elevated miRNAs in endometriotic tissue and/or fluids, five of them (miR-1-3p, miR-122-5p, miR-125b-5p, miR-145-5p and miR-150-5p) are predicted to target progesterone receptors based upon predictive software programs [55,56,57]. Unfortunately, no studies to date have evaluated their function in endometrial or endometriotic tissues and/or cells or their ability to regulate progesterone receptor expression in any tissues. Data on the potential function of these miRNAs can be gleaned from in vitro studies conducted in the field of cancer biology. Table 1 summarizes these five miRNAs, their potential function in cancer and which PGR they may target (based upon bioinformatics). It should be noted that the targeted protein which mediates these processes are not modulated by progesterone. However, these miRNAs putatively may target the progesterone targets in the far right column and this clearly warrants investigation. Interestingly, all of these miRNAs have been reported to decrease cell proliferation, migration and/or invasion as well as to increase apoptosis; cellular events which would restrict endometriosis development and/or progression and are inconsistent with progesterone resistance. Thus, it is tempting to speculate that the increased expression of these miRNAs may be more of a response to disease, attempting to limit ectopic cell survival. The fact that each of these miRNAs are known tumor suppressors which limit cell proliferation, invasion, migration and/or EMT and induce apoptosis would support this notion. 

## 4. Conclusions and Future Directions

Endometriosis is a disease of progesterone insensitivity which is postulated to contribute to persistency of disease and associated symptoms. The altered expression of progesterone receptors including PGR-A/B, mPRs and PGRMCs has been reported in the literature, but the mechanisms which lead to these alterations are poorly understood. Altered miRNA expression in eutopic and ectopic tissue represents one potential mechanism which may contribute to this altered progesterone receptor expression and resulting impaired progesterone responsiveness. From studies conducted to date, miR-29c-3p, miR-126-3p and miR-143-3p have emerged as potentially playing a permissive role in events conducive to endometriosis progression and survival. Unfortunately, these, or any other studies, have yet to evaluate the potential involvement of progesterone receptors in this process. Clearly, future studies are warranted to examine the concurrent regulation of progesterone receptor expression, progesterone resistance and cell proliferation, migration and invasion in response to miR-29c, 126-3p and miR-143-3p upregulation. Equally important will be studies assessing why these miRNAs are misexpressed in endometriosis tissues, which should provide greater insight into the cause and effect relationship between their altered expression and progesterone resistance.

Of those well characterized miRNAs whose expression is elevated in the circulation and/or lesion tissue (miR-1-3p, miR-122-5p, miR-125b-5p, miR-145-5p, and miR-150-5p) and potentially target progesterone receptors, it will be of interest to determine if in fact any of these progesterone receptors are valid targets of these miRNAs. To begin to do so, these miRNAs should be co-localized with their putative progesterone receptor targets in endometriotic lesion tissue cell types to confirm their patterns of expression. Functional studies incorporating 3′ UTR reporter assays confirming binding of miRNA to 3′ UTR as well as over/under-expression of these miRNAs and their impact on cell proliferation, migration, invasion and apoptosis would provide proof of principal. As mentioned earlier, the majority of published studies support the notion that these miRNAs suppress the proliferation, migration and invasion in cancer cells. It is well established that these events are conducive to endometriotic lesion survival, which raises the conundrum of cause and effect. In theory, elevated levels of these miRNAs would suppress progesterone signaling, which would augment lesion survival. This is in clear opposition to the reported outcomes of the aforementioned studies which demonstrated a reduction in the cellular processes conducive to lesion survival. The concurrent assessment of the impact of upregulation of miR-1-3p, miR-122-5p, miR-125b-5p, miR-145-5p, and miR-150-5p on progesterone receptor expression, signaling and cell proliferation, invasion, migration and apoptosis will provide the necessary clues to understanding if these miRNAs play more of a protective role in limiting ectopic lesion survival. As miRNAs are capable of enhancing translation in some cells/tissues, it would be intriguing to decipher if the elevation of these miRNAs leads to an up-regulation of progesterone receptors/progesterone signaling which in turn might suppress the cellular events for lesion survival.

## Figures and Tables

**Figure 1 cells-11-01096-f001:**
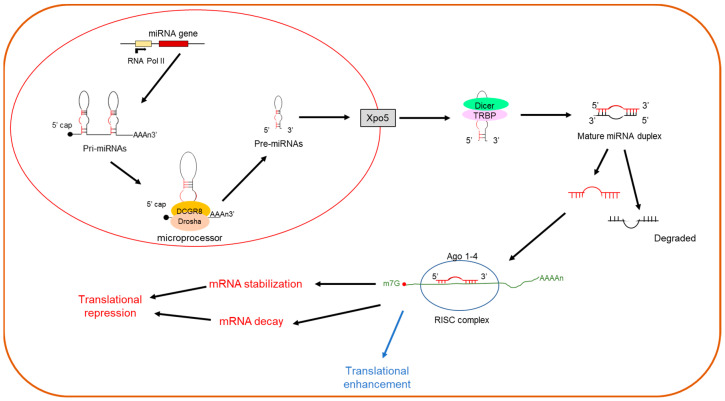
Canonical pathway for miRNA biogenesis.

**Figure 2 cells-11-01096-f002:**
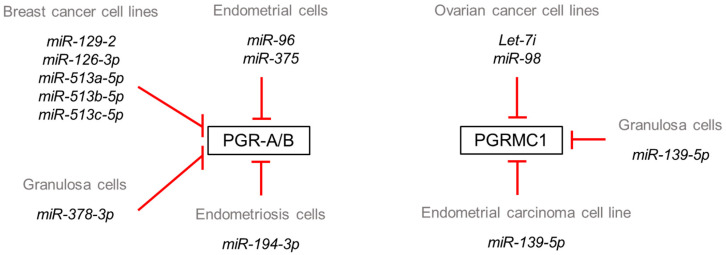
Validated miRNAs which target PGRA/B and/or PGRMC1 in specific cell types.

**Figure 3 cells-11-01096-f003:**
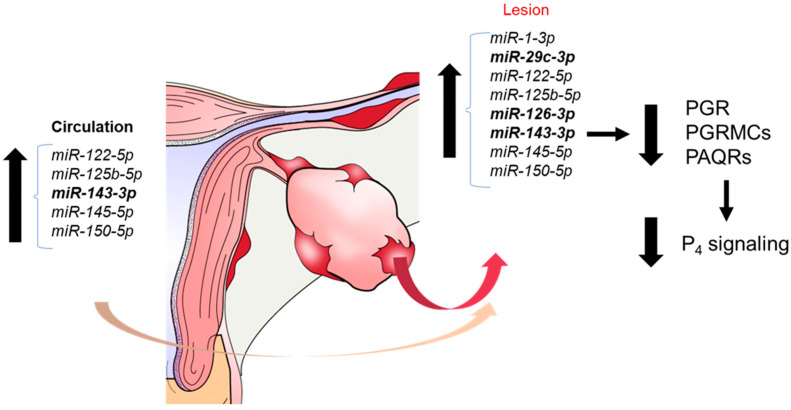
Elevated miRNAs in the circulation and/or lesion of endometriosis patients that may target PGRA/B, PGRMCs or PAQRs.

**Table 1 cells-11-01096-t001:** miRNAs elevated in endometriosis which are predicted to target PGRs which suppress cellular events conducive to endometriosis progression and survival.

miRNA ^1^	Function ^2^	PGR Target ^3^
miR-1-3p [73,74]	Decrease cell proliferation, migration, invasion and increased apoptosis [86,87,88,89,90,91,92]	PGRA/B, PGRMC2, PAQR5, PAQR7
miR-122-5p [93,94]	Decrease cell proliferation, migration and invasion, increased apoptosis [95,96,97]	PGRA/B, PGRMC2, PAQR6, PAQR7
miR-125b-5p [74,84,98,99]	Decreased cell proliferation, migration and invasion [100,101]	PGRA/B, PAQR5, PAQR7
miR-145-5p [73,74,82]	Decreased cell proliferation, migration and invasion, increased apoptosis [102,103,104,105]	PGRA/B, PAQR6,PAQR8
miR-150-5p [82,99]	Decreased cell proliferation, migration and invasion, increased apoptosis [106,107,108]	PGRA/B, PAQR5

^1^ References which reported elevated miRNAs. ^2^ References which reported proposed function. ^3^ Putative PGR targets based upon predictive software programs [55,56,57].

## Data Availability

Not applicable.

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
