# Peer review of "MicroRNAs and Progesterone Receptor Signaling in Endometriosis Pathophysiology"

_cells, 2022, doi:10.3390/cells11071096_

Round 1

Reviewer 1 Report

In the present review, Dr. Nothnick has created an exhaustive summary of the state of microRNA research as it intersects with progesterone signaling and the conundrum of whether they are causative or protective of tissues where endometrial lesions are present.  The initial description of endometriosis complications and hormone dependence in development and treatment schemes sets the stage for the general discussion of progesterone signaling.  This section nicely outlines the classical and non-classical receptors for progesterone signaling and handles well the history and instances where mutation or ablation studies have produced results that are occasionally in conflict with one another.  The general summary of miRNA transcription and processing is more than sufficient for readers to orient themselves and provides a wealth of resources for those wishing to know more beyond the direct links to progesterone signaling.

          I am familiar with the classical PRKO studies in both ovary and uterus and Dr. Pru’s recent work with PGRMCs, but had forgotten that it’s been 20 years since membrane associated progesterone receptors were described in the uterus.  Thus, this review was a good reminder of that prior work that should interest new and established readers working with hormone signaling models.

          Figure 2 is an excellent summary of the bioinformatic work performed by Dr. Nothnick to identify the key miRNA that are later followed by an in-depth discussion of the studies that link them to endometriosis, progesterone signaling, or both.  A minor issue is the legend is misplaced, probably due to formatting imposed by the MDPI template.  The inclusion of only targets with at least two independent laboratories confirming misexpression or upregulation, etc. increases the rigor that the targets that are explored are likely genuinely involved in progesterone resistance or endometriosis.

          Figure 3 is a little difficult to interpret if one wants to be picky.  Is the red arrow coming from a lesion adhered to the ovary?  The circulation is pictured off-screen so it’s not clear from the image where they are acting.

          I like the discussion concerning the upregulation of miRNA may be a natural defense mechanism to protect the peritoneal cavity from unwanted proliferation in the wrong context.

          The overall conclusions are appropriate, and a nice array of proposed future work is presented that makes sense to me.  At least is captured many of the questions that were created in my mind as I read the article.

The overall flow of the article is excellent and only minor mistakes and suggestions for clarity need to be addressed.

Minor issues

The transition of section 2 to section 3 is a bit abrupt.  L227 posits that miRNA could be one cause of altered PGR expression (the title of the article is obviously also a clue).  However, I think the incorporation of some of the background in L255-58 might create a better transition that reads more like a rationale/hypothesis rather than just jumping to the hypothesis.

Mouse nomenclature is inconsistent, particularly in section 2.3.  Not all mouse genes are in italics and proteins in all caps.

Very minor issues

L54, I think a “due” is missing.

L57, extra space after reference 11

L95, “allow” not “allowed”

L147, I think “in normal and endometrial and” is missing something, but I’m not sure if you meant normal and diseased endometrial tissues or something else.

L149, there’s a superfluous “i” in endometriosis

L150, expression

L306, no apostrophe is needed for its, I think.

L329, HOXA10 3’UTR, HOXA10 should be in italics

L335, “have” not “shave”

Author Response

Reviewer 1

Thank you for your insightful comments.  I have replied to each point with my response in bold font.

  1. A minor issue is the legend is misplaced, probably due to formatting imposed by the MDPI template.  I have corrected this but will confirm the placing with the editorial office.
  2. Figure 3 is a little difficult to interpret if one wants to be picky. Is the red arrow coming from a lesion adhered to the ovary?  The circulation is pictured off-screen so it’s not clear from the image where they are acting.  Information this is added to the text.
  3. The transition of section 2 to section 3 is a bit abrupt. L227 posits that miRNA could be one cause of altered PGR expression (the title of the article is obviously also a clue).  However, I think the incorporation of some of the background in L255-58 might create a better transition that reads more like a rationale/hypothesis rather than just jumping to the hypothesis. This paragraph has been-reworded to make a better transition.

  1. Very minor issues. All have been corrected in the text.

L54, I think a “due” is missing.

L57, extra space after reference 11

L95, “allow” not “allowed”

L147, I think “in normal and endometrial and” is missing something, but I’m not sure if you meant normal and diseased endometrial tissues or something else.

L149, there’s a superfluous “i” in endometriosis

L150, expression

L306, no apostrophe is needed for its, I think.

L329, HOXA10 3’UTR, HOXA10 should be in italics

L335, “have” not “shave”

Reviewer 2 Report

Review of “MicroRNAs and progesterone receptor signaling in endometriosis pathophysiology” submitted to Cells

Line 31- the author should cite a more comprehensive endometriosis review here. Ref [1] is very specifically focused on miRNAs

Line 34- again Ref [1] does not seem appropriate here

Line 86- Ref [19] makes no mention as to the localization of PGR isoforms

Line 104- Ref [9] is simply the wrong reference here. This paper does not mention PGR

Line 108- 'compared to disease-free controls'

Lines 145-148- assuming the author meant to write 'normal and endometrial cancer tissues' on line 147, it should be noted that in Ref [37] Sinreih et al. found 'mPRα and mPRβ were localized primarily to cell membranes, while mPRγ was localized in the cytoplasm and/or nucleus' specifically in the normal endometrium, and that localization was different in endometrial cancer. The author should either remove the part about endometrial cancer or describe the differences, or simply rewrite this sentence so that it makes sense

Line 149- 'endometriosisi'?

Lines 150-155- this part is written in a way that is misleading, as the author has used the names of proteins to describe gene transcript levels, and this is inconsistent with the original citation [38]. I recommend that the author use the gene names when describing transcripts, ex: PAQR7 (mPRα) 

Lines 230-244- this entire paragraph is missing citations

Line 257- the author cites their own review as two separate entries in the references, [1] and [54]

Lines 296-299- the statements about [62] and [63] aren’t really connected and make for a confusing run-on sentence

Line 334- should be 'progesterone receptors have'

Table 2 is misleading and needs to be corrected. In the “function” column, all cited papers address the function of microRNAs in cancer, not endometriosis, and often not even cancers that are thought to be hormone driven, such as breast or endometrial. The mechanism of action for the miRNAs in these cancer is not related to progesterone receptors, and instead a result of targeting other proteins such as SFRP1 [88], CCL2 [89], BDNF-TrkB [90], MYC [99], KIAA1522 [102], PI3K/AKT [103], NRAS [104], ARF6 [106,107], SPOCK1 [108], PYCR1 [110]. The author should more clearly identify that these functions are characterized in various cancer types, not endometriosis or endometrium-related disease, and through targets unrelated to progesterone. 

Author Response

Thank you for your insightful comments.  My responses appear in bold font after the reviewer comment.

Line 31- the author should cite a more comprehensive endometriosis review here. Ref [1] is very specifically focused on miRNAs.  More comprehensive review has  been added, replacing the original Ref. 1.

Line 34- again Ref [1] does not seem appropriate here. Corrected.

Line 86- Ref [19] makes no mention as to the localization of PGR isoforms.  Corrected

Line 104- Ref [9] is simply the wrong reference here. This paper does not mention PGR.  I apologize for this error, as this statement is not consistent with the focus of the paragraph, it has been deleted.

Line 108- 'compared to disease-free controls'. Corrected.

Lines 145-148- assuming the author meant to write 'normal and endometrial cancer tissues' on line 147, it should be noted that in Ref [37] Sinreih et al. found 'mPRα and mPRβ were localized primarily to cell membranes, while mPRγ was localized in the cytoplasm and/or nucleus' specifically in the normal endometrium, and that localization was different in endometrial cancer. The author should either remove the part about endometrial cancer or describe the differences, or simply rewrite this sentence so that it makes sense.  Thank you, the sentence has been re-written.

Line 149- 'endometriosisi'?  Corrected.

Lines 150-155- this part is written in a way that is misleading, as the author has used the names of proteins to describe gene transcript levels, and this is inconsistent with the original citation [38]. I recommend that the author use the gene names when describing transcripts, ex: PAQR7 (mPRα).  Corrected.

Lines 230-244- this entire paragraph is missing citations. References added.

Line 257- the author cites their own review as two separate entries in the references, [1] and [54]. Corrected.

Lines 296-299- the statements about [62] and [63] aren’t really connected and make for a confusing run-on sentence.  Corrected.

Line 334- should be 'progesterone receptors have'  Corrected.

Table 2 is misleading and needs to be corrected. In the “function” column, all cited papers address the function of microRNAs in cancer, not endometriosis, and often not even cancers that are thought to be hormone driven, such as breast or endometrial. The mechanism of action for the miRNAs in these cancer is not related to progesterone receptors, and instead a result of targeting other proteins such as SFRP1 [88], CCL2 [89], BDNF-TrkB [90], MYC [99], KIAA1522 [102], PI3K/AKT [103], NRAS [104], ARF6 [106,107], SPOCK1 [108], PYCR1 [110]. The author should more clearly identify that these functions are characterized in various cancer types, not endometriosis or endometrium-related disease, and through targets unrelated to progesterone.  Thank you for this comment, Table 2 has been modified and corresponding text.

Reviewer 3 Report

Endometriosis is a prevalent disorder worldwide and one of frequently examined pathophysiology in the gynecological setting. Understanding its etiology and management is essentially required to derive some better management strategies. Endometriosis is a known progesterone resistance and estradiol-driven phenomenon at the endocrinology level. The author has presented the progesterone receptor and miRNAs based on current development in endometriosis pathogenesis. However, the article seems inappropriate in the present form as the information presented is not either based on miRNAs or progesterone receptor signaling keeping in mind the progress of the same development in the current situation in the literature. There are several concerns where a clear understanding of miRNA in the endometriotic cells; epithelial or stromal types along with the site or grades are lacking. The author needs to emphasize the progesterone receptor signaling activator and downstream signaling targets and the event outcome. The information is given in general and not to a particular topic, title, based also lacks cohesiveness in the miRNAs and progesterone receptors signaling. I may wish to suggest the miRNAs driving the progesterone signaling in the endometriotic cells events in the pathogenesis of the endometriosis in different clinical or model-based settings and current possible clinical or experimental targets based on the miRNA-progesterone or vice versa signaling.

Author Response

Thank you for your insightful comments.  My reply to your concerns are in bold font after the comment.

Endometriosis is a prevalent disorder worldwide and one of frequently examined pathophysiology in the gynecological setting. Understanding its etiology and management is essentially required to derive some better management strategies. Endometriosis is a known progesterone resistance and estradiol-driven phenomenon at the endocrinology level. The author has presented the progesterone receptor and miRNAs based on current development in endometriosis pathogenesis.

However, the article seems inappropriate in the present form as the information presented is not either based on miRNAs or progesterone receptor signaling keeping in mind the progress of the same development in the current situation in the literature. There are several concerns where a clear understanding of miRNA in the endometriotic cells; epithelial or stromal types along with the site or grades are lacking.  I agree with this concern, the majority of the studies assessing endometriotic lesion or even eutopic endometrial tissue have done so on a whole tissue level, without localization.  This point has been added

The author needs to emphasize the progesterone receptor signaling activator and downstream signaling targets and the event outcome. The information is given in general and not to a particular topic, title, based also lacks cohesiveness in the miRNAs and progesterone receptors signaling. I may wish to suggest the miRNAs driving the progesterone signaling in the endometriotic cells events in the pathogenesis of the endometriosis in different clinical or model-based settings and current possible clinical or experimental targets based on the miRNA-progesterone or vice versa signaling.

Round 2

Reviewer 2 Report

The authors corrected the manuscript.  I have no further comments.

Reviewer 3 Report

Thanks for the opportunity to re-review the manuscript. I appreciate that the authors have considered the suggestions to incorporate in the revised version of the manuscript. However, the suggested information is inadequate to consider useful in the revised manuscript.  The given MS title and incomplete information suggest rather compel incorporating the detailed information of the past studies on the miRNA and progesterone receptor signaling to justify the MS discussing the same in the endometriosis biology.